# Lessons Learned from the COVID-19 Pandemic: Interpreting Vaccination Strategies in a Nationwide Demographic Study

**DOI:** 10.3390/vaccines12060581

**Published:** 2024-05-26

**Authors:** Igor Age Kos, Faissal Nemer Hajar, Gustavo Sarot Pereira da Cunha, Claudia Corte, Luisa Augusto Furlan, André Santa Maria, Douglas Valverde, Bárbara Emoingt Furtado, Miguel Morita Fernandes-Silva, Valderilio Feijó Azevedo

**Affiliations:** 1Edumed Educação em Saúde, 2495 Bispo Dom José Street, Curitiba 80440-080, Brazil; igor.kos@uks.eu (I.A.K.);; 2Department of Internal Medicine I, Hematology, Oncology and Rheumatology, Saarland University Medical Center, Kirrbergstr. 100, 66424 Homburg, Germany; 3AstraZeneca Brazil, Raposo Tavares Road, KM 26.9, Cotia 06709-000, Brazil; 4Techtrials, Dr. Ovídio Pires de Campos Street, São Paulo 05403-010, Brazil; 5Department of Internal Medicine, Federal University of Paraná, Rua General Carneiro, 181, 10° Andar, Alto da Glória, Curitiba 80060-900, Brazil

**Keywords:** vaccination, COVID-19, Brazil, SARS-CoV-2

## Abstract

Objective: Brazil was strongly affected by the COVID-19 pandemic. Its continental dimension and socio-demographic characteristics pose challenges to distribution and accessibility, making vaccination programs challenging. The objectives of the study were to describe the clinical and demographic characteristics of the general population vaccinated against COVID-19 by October 2021 and analyze the strategies implemented during the vaccination program. Study design and setting: A retrospective nationwide study that analyzed data from the OpenDataSUS platform of the Informatics Department of the Brazilian Ministry of Health (DataSUS), which contains information from all individuals in Brazil who have received at least one dose of any vaccine against COVID-19 approved by the National Health Agency (ANVISA) from 17 January to 3 October 2021. Results: Until 3 October, a total of 146,254,578 persons (68.6 per 100 inhabitants) received at least one dose of a vaccine in Brazil. The north and northeast regions had the lowest vaccination rates compared with the remaining regions (North: 56.8, Northeast: 62.0, South: 74.4, and Southeast: 73.2 per 100 inhabitants). Elderly individuals had the highest vaccination rates, particularly those above 70 years old. Heterologous dosing regimens were administered to 1,063,079 individuals (0.7% of those receiving the first dose). Conclusions: The COVID-19 vaccination program reached more than two-thirds of the population in Brazil by 9 months after its start, but the vaccination coverage was heterogeneous, reflecting the country’s geographic and socio-demographic characteristics. Establishing priority groups for vaccination was a main characteristic of the vaccination strategy. In addition, technology transfer agreements have played an important role in increasing vaccine accessibility.

## 1. Introduction

By October 2021, the primary coronavirus disease 2019 (COVID-19) vaccination deployment was finalized in Brazil. At that time, the disease caused by SARS-CoV-2 had been reported in over 240 million people worldwide, leading to approximately 4.9 million deaths [1]. Brazil had reported approximately 21 million cases and over 600,000 deaths [2]. Since the pandemic began, uncountable efforts have been invested in order to develop or repurpose efficient therapies against COVID-19 [3]. Mass vaccination programs are one of the most effective public health interventions, since they can reduce mortality and morbidity associated with the disease with possible implications in transmissibility [4,5]. Hence, the development and manufacturing of vaccines became a high priority among biomedical companies worldwide.

In December 2020, the first vaccine against COVID-19 was authorized for emergency use in Europe and the USA, and other countries approved several other vaccines [6,7,8]. To overcome challenges in manufacturing and distribution, countries and companies implemented various strategies [9]. However, other challenges remain, such as medical contraindications, geographical and social constraints, shortage of inputs, and individual choice [5,10,11,12].

Brazil is the fifth largest and sixth most-populated country in the world, with 213 million inhabitants. Despite its high gross domestic product, Brazil is one of the most unequal countries [13,14]. The Amazon rainforest encompasses a total of 5,500,000 km^2^, of which approximately 60% is located in Brazil [15]. Therefore, the country faces vaccine-related challenges due to its size and resources. Nevertheless, the Brazilian healthcare system has managed successful vaccination plans in the last few decades, with the first one being created in 1973 [16]. Currently, 20 vaccines are regularly given to individuals of different ages [16].

Four different vaccines against COVID-19 were approved by the Brazilian Health Regulatory Agency (ANVISA) at the time of data collection: Comirnaty (Pfizer/Wyeth), ChAdOx1 nCov19, CoronaVac (Sinovac-Butantan partnership), and Janssen vaccine (Janssen-Cilag). The Comirnaty and ChAdOx1 nCov19 vaccines have full market authorization, whereas the latter two were granted emergency use approvals at the beginning of 2021 [17]. CoronaVac was approved after a phase 3 trial was performed in Brazil following a technology partnership between the Brazilian Butantan Biomedical Center and the Chinese company Sinovac Biotech. Furthermore, in June of 2021, a technology transfer was signed between the Immunobiological Technology Institute (Biomanguinhos) of the Oswaldo Cruz Foundation (FioCruz) and the company AstraZeneca, which allowed for the national production of the vaccine [18,19].

This study aimed to analyze the clinical and demographic characteristics of the Brazilian population vaccinated against COVID-19 in 2021 up to October 2021, examining the population characteristics according to product, vaccine intervals, and the number of doses.

## 2. Materials and Methods

### 2.1. Study Design and Populations

We performed a retrospective descriptive study using data from a public healthcare database in Brazil. COVID-19 vaccines were only available in the Brazilian public health care system, where vaccination reports are mandatory. Records of vaccination were obtained from all persons living in Brazil who received at least one dose of one of the four vaccines against COVID-19 approved by ANVISA from 17 January 2021 to 3 October 2021. We collected data on age, sex, race, manufacturer, date of administration, and geographic region for this analysis.

The study population was further stratified into the following subgroups for analysis and comparisons: complete standard vaccination, heterologous dosing regimen, three or more doses, and delayed vaccination. The first group comprised individuals who received one dose of the Janssen vaccine or two doses of a vaccine from the same manufacturer; the heterologous dosing regimen group included persons who received the first two vaccine doses from different manufacturers (except for Janssen-Cilag) as part of a two-dose primary series; three or more doses included individuals who received three or more vaccines irrespective of manufacturer and dose interval. Due to the official recommendations of the Brazilian Ministry of Health authorizing a booster dose for persons over 70 years and immunocompromised individuals, starting on 15 September 2021, we performed a separate analysis of the third dose of vaccines that were administered until this date, in order to identify deviations from the official schedule. Finally, the delayed vaccination group comprised individuals receiving a delayed administration of the second vaccine dose (among vaccines with a two-dose primary series), i.e., later than recommended by the manufacturer as detailed below. Persons whose second vaccine dose was due after the period of data collection were not included in this analysis. Persons with a register of only one vaccination, with the exception of Janssen-Cilag, were analyzed separately. Each subgroup was compared with the complete standard vaccination group to establish differences regarding key variables.

Individuals with incomplete vaccinations were defined as those receiving one dose of a vaccine product (excluding Janssen-Cilag) and without records of a second vaccination until the data cutoff. Individuals whose second dose was planned after the data cutoff were excluded.

Persons with incomplete data regarding the key variables of age, sex, vaccination date, and manufacturer, as well as persons with incorrect data, specifically negative vaccine intervals, were excluded from the analysis. Vaccination according to self-informed race as per the Brazilian Institute for Geography and Statistics (IBGE) was also included in the analysis as follows: black, brown, white, East Asian, or Indigenous (in Portuguese: preta, parda, branca, amarela ou indígena). Of note, brown is a free translation of the word “pardo”, which is used by IBGE for self-reporting of race. Because of miscegenation, many individuals do not recognize themselves as any of the other colors/races mentioned. “Pardo” or brown was the most common self-reported race in Brazil in the year of 2022 [20].

### 2.2. Available Vaccines and Intervals

At the time of the study, four vaccines were available for distribution in Brazil. Three of them were based on a two-dose regimen, Cominarty (Pfizer/Wyeth), ChAdOx1 nCov-19 (AstraZeneca/FioCruz/SII), and CoronaVac (Sinovac/Butantan), while the Janssen vaccine (Janssen-Cilag) provided complete immunization after one single shot. It is important to highlight that both the National Immunization Program and the Informatics Department of the Brazilian Ministry of Health (DataSUS) grouped all ChAdOx1 nCov-19 vaccines under the same codification, although 3 different sources were available until the study data cutoff: Covax Facilities (Italy and South Corea), Fiocruz-AstraZeneca partnership (Brazil), and Serum Institute (India). The following intervals were considered “as scheduled”: 90 days (acceptable range 83–97) for Comirnaty, which was due to orientations of the Brazilian Ministry of Health to increase the vaccination interval initially to 8 weeks and later to 12 weeks. For ChAdOx1 nCov-19, the same interval was applied. The expected interval for CoronaVac was 28 days (acceptable range 25–31). For individuals receiving heterologous vaccinations, which were not officially recommended by the Brazilian Ministry of Health, the expected interval was based on the recommendations of the manufacturer of the first shot, with the exception of those receiving Janssen-Cilag as the first manufacturer.

### 2.3. Data Collection and Statistical Analysis

All data were sourced from the DataSUS, and more specifically from the OpenDataSUS platform. To establish comparisons and present data relative to the complete Brazilian population, we used projections for the year 2021 of census data based on Censo 2010, the last official population census performed in Brazil. Vaccination records and variables of interest were then uploaded into a second database using Microsoft Power BI.

Discrete variables were presented as number (N) and percentage (%). The socio-demographic characteristics of individuals who received doses from different vaccine manufacturers were compared to those who received two doses from the same vaccine manufacturer. Similarly, the characteristics of individuals with the delayed administration of the second dose were compared with those who took the second dose within the recommended interval. The mean time of delay between the first and second dose and standard deviation were displayed within the specific subgroups. The mean time between doses and standard deviation as well as the socio-demographic characteristics of interest were displayed for individuals receiving three or more doses until 15 September 2021.

## 3. Results

### 3.1. General Results

Until 3 October 2021, a total of 146,254,578 persons received at least one dose of the vaccine in Brazil, corresponding to 68.6 per 100 inhabitants according to the census data projection for 2021. Considering the different Brazilian regions, north and northeast Brazil had the lowest vaccination rates, 56.8 and 62.0 per 100 inhabitants, respectively, whereas south and southeast achieved higher rates of vaccination of 74.4 and 73.2 per 100 inhabitants, respectively (Figure 1). The center-west region vaccinated 69.7 persons per 100 inhabitants. Table 1 summarizes the vaccination rates according to the geographic regions and key parameters. Figure 2 shows the cumulative number of patients that received each dose during the months.

Elderly individuals had the highest vaccination rates, particularly those over 70 years old. Adults between 20 and 60 years old had heterogeneous rates, with lower rates in the north and northeast (Table 1). Even though there was virtually no vaccination amongst children under 9 years old, there were still records of vaccination amongst these age group. Of note, vaccination of people under 18 years old was not yet recommended at this time. The subgroup between 10 and 14 years had the lowest coverage (16.3 per 100 inhabitants). The north, northeast and center-west regions had higher rates in individuals over 90 years old (96.8, 103.1, and 96.9 per 100 habitants, respectively) compared to the south and southeast (76.8 and 80.8 per 100 habitants, respectively).

The AstraZeneca/Fiocruz/SII-manufactured vaccine was the most frequently administered vaccine product with 29.7 per 100 inhabitants, followed by the Pfizer-manufactured vaccine with 20.2, Sinovac/Butantan with 19.1, and Jansen-Cilag with 2.2 per 100 inhabitants.

### 3.2. Complete Regimens and Heterologous Dosing Regimen

Of the population receiving complete vaccination regimens (79,790,576), 78,727,497 (98.7%) persons received two doses from the same manufacturer, whereas 1,063,079 (1.3%) were administered with vaccines from different manufacturers (Table 2). Higher proportions of heterologous dosing regimens were observed amongst middle aged groups (40–59), peaking in the age group between 50 and 54 years, and falling clearly in populations of 60 years old or more.

The southeast region accounted for most of the individuals with complete vaccinations from the same manufacturer (46%), followed by the northeast (22.7%) and south (17%). Similarly, the southeast region presented the highest percentages of persons with heterologous dosing regimens (75.6%), followed by the northeast region (11.7%) and center-west (5.3%). The south region had the second lowest proportion of heterologous dosing regimens amongst regions with 4.5%, whereas the north region had the lowest percentages of complete vaccination with both homologous and heterologous dosing regimens (6.4% and 3%, respectively) (Appendix A).

AstraZeneca/FioCruz/SII was the most administered homologous vaccination (47.9%), followed by Pfizer (11.5%), and Sinovac/Butantan (8.6%) (Figure 3). Amongst individuals receiving a heterologous dosing regimen, AstraZeneca/FioCruz/SII was the most frequent manufacturer administered as the first dose (88%), followed by Sinovac/Butantan (8.6%) and Pfizer (3.4%).

### 3.3. Incomplete Vaccination and Delayed Administration

In general, incomplete vaccination—as described in the Methods section—was reported in 15,181,376 persons with a rate of 7.1 per 100 inhabitants. Higher rates were observed in the southeast region (8.2) and lower in the south region (4.0) (Appendix A).

Incomplete vaccinations were found in 16.2% of individuals receiving the first dose from AstraZeneca/FioCruz/SII, 14.6% from Sinovac/Butantan, and 19.2% from Pfizer. Higher rates of incomplete vaccination were mostly observed in younger populations, especially between 15 and 54 years old. In addition, Indigenous, brown, and East Asian populations were most likely to have an incomplete vaccine regimen.

Among 79,790,576 (37.3 per 100 inhabitants) individuals receiving two doses, the second dose was administered with a delay in 37.5% of them. Table 2 summarizes the data regarding delayed administration. According to each region, the southeast and northeast presented the highest percentages of delayed administration (41.5 and 23.5%, respectively). However, proportionally to administration as scheduled, the north, northeast, and center-west were more frequently delayed.

When stratified by race, Indigenous and East Asian persons were most likely to receive a delayed administration. In addition, older populations, especially between 65 and 79 years, had higher rates of delay. Sinovac/Butantan was the most frequent manufacturer amongst individuals with a delayed second dose (52.4%), followed by AstraZeneca/FioCruz/SII (37.7%) and Pfizer (12.8%).

### 3.4. Three or More Doses

In total, 194 009 persons received a third dose of the vaccine by 15 September 2021, before recommendations were in place for a third-dose booster. The region with the most administered three doses of vaccines was the southeast (65.2%) followed by center-west (18.1%). The north and south regions reported the smaller percentages of a third dose (0.7 and 5.4%, respectively). Older populations, especially those older than 70 years, received most frequently a third dose of the vaccine, with higher proportions amongst those older than 85 years. According to race, white persons received the highest percentage of a third dose (43.6%). When stratified by manufacturer, 86.5% of persons receiving a third dose had Sinovac/Butantan as the manufacturer of the first dose, compared to 2.7% and 0.2% receiving AstraZeneca/FioCruz/SII and Pfizer, respectively. Pfizer was the most frequently administered as the third dose, followed by Sinovac/Butantan.

## 4. Discussion

In this study, we described the COVID-19 vaccination coverage in Brazil. By 3 October 2021, the vaccination rates were high among the elderly, with virtually all individuals between 60 and 89 years being vaccinated, reflecting the prioritization strategy adopted by the national immunization program [22]. We found significant heterogeneity in the vaccine distribution across geographic regions. Vaccination rates were higher in the south and southeast, compared to the north and northeast regions, particularly among adult individuals (20 to 60 years old). Among individuals receiving two doses of the vaccine, almost 40% received the second dose later than originally scheduled, with higher proportions of delayed doses in the north and northeast regions. Overall, the most common vaccine was that manufactured by AztraZeneca, followed by Pfizer, and Sinovac/Butantan. The former was predominant in the adult age range, while the latter was preponderant among the elderly older than 65 years old. According to the national immunization program, the priority group for COVID-19 vaccination was the elderly population. Sinovac-Butantan was the first vaccine available in Brazil, and because of that, it was used first in the priority groups, followed by AstraZeneca, which was available a few weeks after Sinovac.

Brazil started its COVID-19 vaccination program when it had the third-highest number of cases worldwide. Early start and fast vaccine delivery are important to reduce deaths and hospitalizations [23,24]. Vaccination in Brazil began a month later than in other countries, yet by October 3rd it had already surpassed the vaccination coverage of China, the USA, and Germany [1].

By understanding the temporal development of the vaccination program, strategies can be developed to improve vaccination coverage. Since, in the country, the COVID-19 vaccination was exclusively administered through the public health service, our analysis comprises virtually the totality of data generated in the country.

The heterogeneous vaccination coverage in Brazil cannot be explained by delays in the start of the vaccination. All Brazilian states administered their first shots of vaccination almost simultaneously: between 17 January (São Paulo) and 19 January (Distrito Federal and other 10 states) [25]. The Brazilian program initially planned to equally distribute the vaccine shots among states according to their populations; however, the state of Amazonas, in the north region, was prioritized at the beginning of the immunization program, due to its high number of cases and the emergence of a new variant during the first months of 2021 [26]. Despite that, vaccination rates in the north region were slower. The north and northeast regions have the lowest urbanization rates in the country, which may impose challenges to vaccine access, distribution, and storage [27]. In addition, the north region has the largest population of Indigenous peoples, who often live in difficult-to-reach locations [28]. The Amazon rainforest encompasses all states in the north region and most of the land is covered by it. This also imposes logistical challenges to the storage and distribution of vaccines, whose molecular stability depends on an efficient cold chain.

Other reasons, beyond logistics in production and distribution, may also explain this heterogeneity in the vaccination program. There are considerable differences in the Human Development Index (HDI) among the Brazilian regions. The lowest vaccination rates were found in the states with lower HDI: north 0.667, northeast 0.663, center-west 0.757, southeast 0.766, and south 0.754 [29].

The distribution of the vaccine manufacturers reflects many aspects of the vaccination program in Brazil. AstraZeneca/FioCruz/SII and Sinovac/Butantan were the first manufacturers to obtain emergency approval, which occurred in early January 2021. The Brazilian immunization program started after importing six million shots of the Sinovac/Butantan and two million shots of the AstraZeneca/FioCruz/SII vaccine [30,31]. On 8 March 2021, the AstraZeneca/FioCruz/SII vaccine began to be produced nationally, following a technology transfer agreement signed between AstraZeneca and FioCruz [32]. Figure 4 illustrates the clear increase in the vaccination numbers with the AstraZeneca/FioCruz/SII product after this date, further leading the product to be the most administered in the vaccination program according to this data.

Similarly, the Butantan Institute developed a partnership with the Chinese company Sinovac, which led to the completion of a phase III study and allowed for the national production of the vaccine [33,34]. This technology transfer agreement played an important role in the distribution and availability of vaccination. The Sinovac/Butantan vaccine was the most administered manufacturer during the first phase of vaccination and remained the base of the program in some states, such as São Paulo. Put together, these data highlight the importance of such strategies to increase local production, which enhances the country’s autonomy. In Brazil, this process was facilitated through its long-lasting experience in producing biological products, including vaccines [35]. Of note, even though transfer agreements may increase speed and distribution, it is important to point out that many of the inputs required are probably not locally sourced, and therefore, importation issues and input shortages can also impact production and availability. Table 3 summarizes the challenges and successes of the vaccination program.

Challenges due to vaccine shortages may have resulted in delayed administration of the second shot, particularly due to governmental decisions. Two major delays in the importation and delivery of inputs and the withdrawal of 25 batches of Sinovac/Butantan produced in a non-inspected production site may have negatively impacted the rates of administration and compliance of the second dose [31,36,37,38,39]. This may explain why second shots were more frequent among older individuals, as they were the first ones to be vaccinated.

As the official recommendation of the government for a booster dose (third dose) was only made on 15 September 2021, the groups that received a third dose in our analysis potentially reflect deviations from the immunization program, although we cannot specify if this was due to error or to individual choice. Sinovac/Butantan was the most applied manufacturer of the first dose among those individuals with three doses. This was the most available product at the beginning of the program and required shorter intervals between doses, meaning that persons that were immunized with this vaccine were eligible earlier for a booster shot.

Our study has limitations that deserve attention, such as data inconsistencies due to the informatization process starting late, record errors, wrong administration or reporting after international vaccination, and imprecision in population demographic information from the 2010 Census. This could explain the lower vaccination rates among nonagenarians in the southeast and south regions, which may be due to an overestimation of population size. Furthermore, we are not able to evaluate vaccination coverage within distinct urban zones characterized by a higher population density. Such locales may need expanded coverage and prioritization strategies to mitigate localized outbreaks.

## 5. Conclusions

In conclusion, despite its later start, the COVID-19 vaccination program in Brazil achieved broad vaccination coverage after 9 months, comparable to other countries that started earlier. With the heterogeneous distribution of vaccination coverage resulting from its geographic and socio-demographic characteristics, the 50 years of experience of the Brazilian immunization program, along with technology transfer agreements and high population vaccine acceptance may have contributed to COVID-19 vaccination in Brazil.

## Figures and Tables

**Figure 1 vaccines-12-00581-f001:**
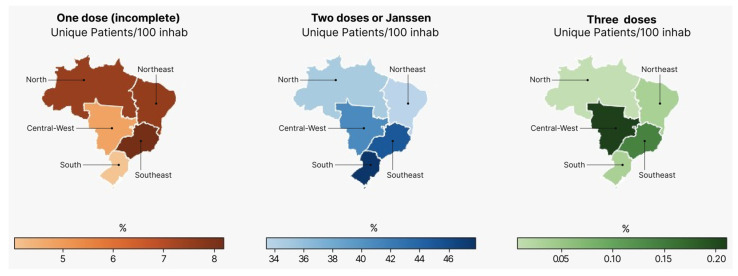
Vaccination uptake by Brazilian region. This figure shows the regional differences between dosing phases across Brazil, normalized by 100 inhabitants. Populational density per region is as follows (2010): Brazil: 22.43 inhabitants/km^2^; North: 3.35 inhabitants/km^2^; Northeast: 30.69 inhabitants/km^2^; Southeast: 86.92 inhabitants/km^2^; South: 48.58 inhabitants/km^2^; Center-west: 8.75 inhabitants/km^2^ [21].

**Figure 2 vaccines-12-00581-f002:**
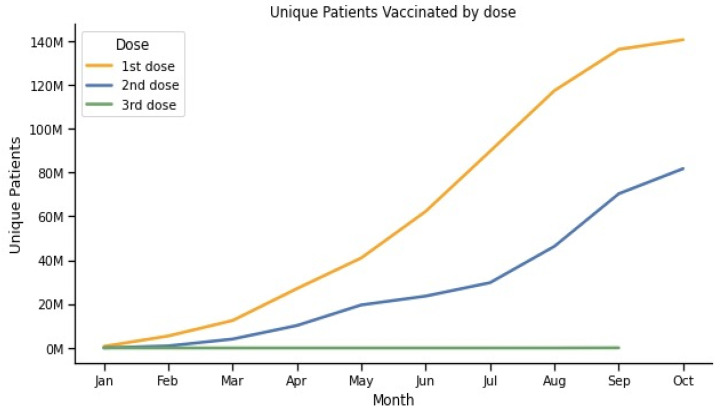
Unique patients vaccinated by number of doses. In this figure, a time series graph shows the cumulative number of unique patients that received each vaccine dose by month.

**Figure 3 vaccines-12-00581-f003:**
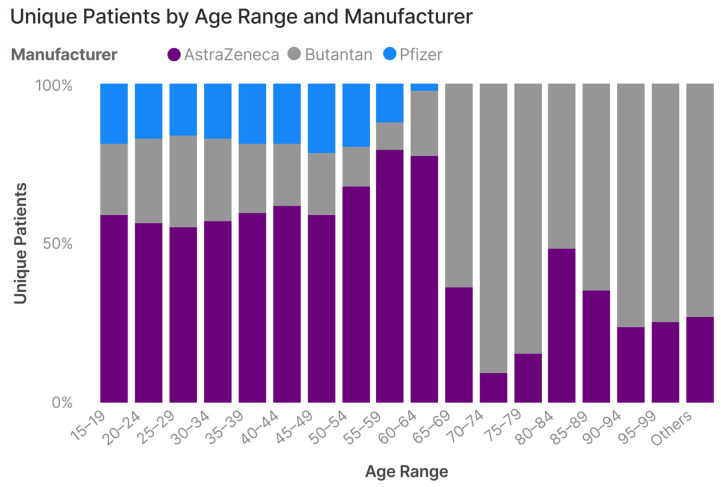
Proportion of vaccination in individuals receiving two shots of vaccine from the same manufacturer. This figure explores the proportion of unique patients who received two shots of vaccine from the same manufacturer across different age groups.

**Figure 4 vaccines-12-00581-f004:**
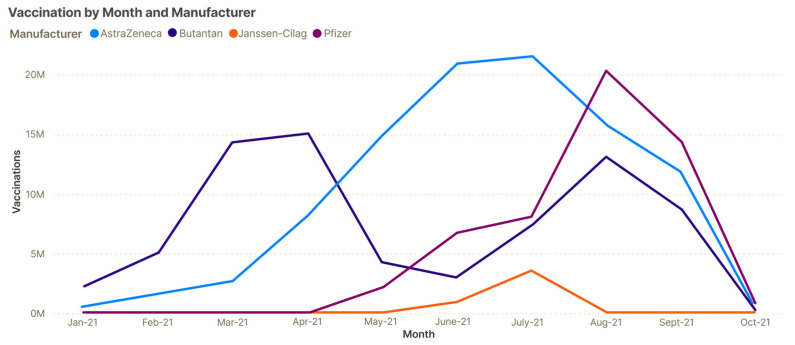
Vaccinations over time according to the manufacturer. The figure plots the total vaccinations grouped by month and manufacturer from January 2021 to October 2021.

**Table 1 vaccines-12-00581-t001:** Proportion of individuals vaccinated between 17 January and 3 October, according to key parameters and geographic region.

Parameter	North	Northeast	Center-West	Southeast	South	All Regions (Brazil)
Vaccinated	10,734,530	35,749,128	11,643,127	65,568,854	22,628,205	146,254,578
Total Pop (census projection)	18,906,962	57,667,842	16,707,336	89,632,912	30,402,587	213,317,639
Percentage vaccinated	56.78	61.99	69.69	73.15	74.43	68.56
Sex (per 100 hab)						
Men	54.31	59.21	67.28	70.70	72.05	66.02
Women	59.26	64.61	72.06	75.49	76.72	70.99
Age category (per 100 hab)						
0–4	0.01	0.02	0.00	0.01	0.01	0.01
5–9	0.01	0.03	0.01	0.02	0.02	0.02
10–14	26.15	15.88	20.95	14.53	10.98	16.31
15–19	55.29	48.88	62.96	64.66	59.69	58.11
20–24	65.22	65.90	81.23	84.05	87.71	77.07
25–29	67.53	71.55	83.92	86.03	90.27	80.55
30–34	68.77	75.51	83.48	86.39	92.59	82.38
35–39	75.54	81.61	88.41	91.22	96.12	87.68
40–44	78.52	83.46	91.58	94.56	97.59	90.42
45–49	80.36	84.70	93.38	95.66	97.55	91.65
50–54	83.61	86.49	96.08	97.28	98.97	93.65
55–59	89.36	91.15	99.52	100.44	101.37	97.47
60–64	95.99	98.77	102.66	102.44	103.17	101.25
65–69	97.15	99.16	103.36	101.97	101.66	101.03
70–74	97.35	98.85	101.96	97.87	98.28	98.37
75–79	97.18	97.42	101.85	97.37	96.21	97.41
80–84	104.42	103.70	106.07	100.55	97.18	101.25
85–89	104.77	107.51	107.75	101.57	96.04	102.51
90+	96.77	103.11	96.93	80.83	76.75	88.09
Race (per 100 hab) *						
Black	24.28	33.97	42.13	45.73	58.06	40.74
Brown	36.26	32.16	41.83	30.91	22.73	32.61
Indigenous	35.64	50.28	40.25	31.57	37.07	39.56
White	26.70	32.73	48.90	50.90	67.88	50.66
Manufacturer (per 100 hab)						
AstraZeneca/FioCruz/SII	26.87	27.09	29.69	30.95	33.1	29.7
Sinovac/Butantan	12.32	16.83	17.04	22.57	18.48	19.09
Janssen-Cilag	1.43	1.55	3.11	2.26	2.85	2.15
Pfizer	18.26	18.76	24.11	19.98	22.76	20.22
Complete primary series						
Two doses (per 100 hab)	26.55	30.99	37.03	41.61	44.14	37.40
Homologous dose regimen	26.38	30.77	36.69	40.71	43.98	36.91
Heterologous dose regimen (2 doses)	0.17	0.22	0.34	0.90	0.16	0.50
Only priming dose (incomplete) (per 100 hab)	7.50	7.59	4.82	8.20	4.05	7.12
2 shots/Janssen (complete) (per 100 hab)	27.98	32.51	39.93	43.72	46.95	39.47
3 shots (per 100 hab)— until 15 September	0.01	0.04	0.21	0.14	0.03	0.09
Specific groups (absolute number of vaccinated individuals) ^a^						
Indigenous population	142,229	105,468	66,136	31,846	18,874	364,458
Comorbidity	700,604	2,658,137	912,614	6,217,896	1,900,805	12,389,602
Healthcare professionals	569,409	6,291,659	1,273,855	5,669,635	1,999,103	15,798,754

* exact data for “East Asian” population not available. ^a^ There were no reliable data on the estimated total population size for these specific groups at the time of this analysis.

**Table 2 vaccines-12-00581-t002:** Vaccination rates according to key parameters in individuals with delayed second doses or with vaccinations as scheduled.

Parameter	Two Doses as Scheduled ^a^	Delayed Dose ^a^
Vaccinated	49,813,650	29,976,926
Vaccinated (per 100 hab)	23.35	14.05
Male sex (% of total vaccinated)	44.91	44.14
Age category (% of total vaccinated)		
0–4	0.00	0.00
5–9	0.00	0.00
10–14	0.00	0.12
15–19	1.43	1.77
20–24	5.10	5.21
25–29	6.28	6.84
30–34	7.56	8.13
35–39	7.36	8.88
40–44	8.97	8.54
45–49	10.50	8.41
50–54	11.95	8.74
55–59	12.23	8.76
60–64	11.74	8.56
65–69	7.47	10.25
70–74	4.92	8.32
75–79	3.31	5.36
80–84	2.71	2.66
85–89	1.44	1.50
90+	0.71	0.97
Race (% of total vaccinated)		
Black	4.48	4.16
Brown	18.28	19.78
Indigenous	0.17	0.55
White	38.36	36.71
Manufacturer of 1st dose (% of total vaccinated)		
Sinovac/Butantan	33.08	52.39
AstraZeneca/FioCruz/SII	55.21	37.67
Pfizer	14.30	12.82
Specific groups (% of total vaccinated)		
Indigenous population	0.17	0.66
Health workers	12.37	15.38
Region of the country (% of total vaccinated)		
North	4.36	9.51
Northeast	21.72	23.52
Center-West	7.19	8.68
Southeast	50.06	41.49
South	16.77	16.91

^a^ As scheduled: vaccination interval according to the recommendations of the Brazilian Ministry of Health described above. Delayed: vaccination interval above the acceptable range according to the recommendations of the Brazilian Ministry of Health described above.

**Table 3 vaccines-12-00581-t003:** Lessons learned from COVID-19 vaccination in Brazil.

Successes	Challenges and Possible Solutions
Technology transfer agreements increased the speed and distribution of vaccines in the country	Difficulty in overcoming input shortage → increase local production of fundamental resources for biologicals
Prioritizing groups at higher risk of developing a severe course of disease was efficient across all regions	Uneven coverage in different regions, with lower vaccination especially for adults in the north and northeast regions → increase access and logistical support for storage and transportation of vaccines. Strategies may have to prioritize certain regions over others because of logistical difficulties, higher incidence of disease, new viral variants, or higher populational density with higher risk of local outbreaks
Heterologous vaccination was relatively low; however, it may have helped to reduce delays in second-dose vaccination	Slow initiation of vaccination and lower availability of products at the beginning of the program → faster negotiation/implementation of technology transfer agreements
Rapid catch-up of coverage (especially due to technology transfer) despite the later start of vaccination	

## Data Availability

The data presented in this study are available in this article and Appendix A.

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
