# Peer review of "Lessons Learned from the COVID-19 Pandemic: Interpreting Vaccination Strategies in a Nationwide Demographic Study"

_vaccines, 2024, doi:10.3390/vaccines12060581_

Round 1

Reviewer 1 Report

Comments and Suggestions for Authors

This manuscript deals with one-year report of COVID vaccines in Brazilian population, looking for four vaccines, being two locally processed from bulk to vaccine and two processed elsewhere as vaccine. The authors, probably due to their affiliation, poorly explore this aspect but if a table sorting by locally produced vs elsewhere produced vaccine was build, probably the efficiency of local vaccine production could be much more interesting and significant.

Brazil is a subcontinental country like US, Canada, Russia or China. There are either areas with low population density or areas densely habited. The vaccine coverage ratio depends in people access to transport to the vaccine office. The authors do not explore it, despite these data are more associated with vaccination failures, especially in North Region, a region with more river or aerial transportation than road or train transport.

Data on age or race are complicated in high miscegenation in the population as Brazil. People identify themselves as white, black or “brown (? Miscegenation?)”, but this a personal or individual opinion not ethnic or race definition. Rarely, a pure genomic ethnic background is identified in a third generation Brazilian citizen, regardless their external aspect. Please explain the brown ethny or remove then from the text. 

The same applies to indigenous population. Usually, Brazilian indigenous people in littoral areas identifies themselves as blacks or “browns??” than Indians to avoid more discrimination, as indigenous people are considered less reliable than other ethnos, where others with high miscegenation and only with a little indigenous background must identify themselves as Indigenous to achieve public benefits.  

Data are presented in long tables without statistical analysis, resulting in difficult reading and comprehension. Those tables could be easily resumed if variables that are much segmented as age groups could be transported to presentation as figures. Figures are also a problem. Maps without population density are not conclusive. When looking the Brazilian maps, North region is the large but with the smaller population. Maps without adequate information only obscure the data significance.

The authors must focused their results in less tables and statistical analysis, not looking for the regional or ethnic differences but in local or elsewhere vaccine production, which is much more important for a subcontinental country as Brazil. 

Author Response

Dear Reviewers, thank you very much for your careful and productive comments regarding our manuscript. Please find a point-by-point answer to your questions and suggestions below. If feel that our answers are still not complete enough, or if there are still open questions, we would be more than willing to discuss these again.

REVIEWER 1

This manuscript deals with one-year report of COVID vaccines in Brazilian population, looking for four vaccines, being two locally processed from bulk to vaccine and two processed elsewhere as vaccine. The authors, probably due to their affiliation, poorly explore this aspect but if a table sorting by locally produced vs elsewhere produced vaccine was build, probably the efficiency of local vaccine production could be much more interesting and significant.

Thank you for this relevant comment. Indeed, locally manufactured vaccines were crucial to increase coverage in the country. As suggested by both reviewers, we have tried to reduce the number of tables, as some of the data was difficult to read and interpret. Thus, we were reluctant to add one more table to the manuscript. As we believe that both Figures 3 and 4 are good representatives of the role of locally produced products, we have added a paragraph in the discussion highlighting how the distribution of vaccines changed dramatically after the technology for manufacturing was available in the country. This can be especially seen in the steep slope from both products after starting official production in Brazil (e.g. March for AstraZeneca/Biomanguinhos product). In addition, figure 1 shows how the majority of first-dose vaccines were administered with products manufactured locally.

Brazil is a subcontinental country like US, Canada, Russia or China. There are either areas with low population density or areas densely habited. The vaccine coverage ratio depends in people access to transport to the vaccine office. The authors do not explore it, despite these data are more associated with vaccination failures, especially in North Region, a region with more river or aerial transportation than road or train transport.

We thank the reviewer for this important remark. As the data show, the north and northeast regions had lower vaccine coverage, especially amongst younger adults. This can be well observed in Figure 1. To further address the challenges of these specific regions we have included elements in the introduction and discussion, explaining the geographical difficulties of these populations, especially the extension of the Amazon rainforest and socio-economical parameters that may help with the interpretation, especially of international readers that are not familiar with Brazilian geographical singularities.

Data on age or race are complicated in high miscegenation in the population of Brazil. People identify themselves as white, black, or “brown (? Miscegenation?)”, but this is a personal or individual opinion not ethnic or race definition. Rarely, a pure genomic ethnic background is identified in a third-generation Brazilian citizen, regardless their external aspect. Please explain the brown ethny or remove then from the text. 

Matters of race are indeed complex and we thank the reviewer for pointing out this topic. We have used the racial definitions according to the Brazilian Institute for Geography and Statistics (IBGE), in which people declare to which race they most identify with, from the following possible: black, brown, white, yellow, or indigenous (in Portuguese: preta, parda, branca, amarela ou indígena). There is no genomic correlation or information to this classification, and it is in line with the government recommendations. Brown is a free translation of the word “pardo”, which is used by IBGE for self-reporting of race. Because of miscegenation, many individuals do not recognize themselves as any of the other colors/races mentioned. “Pardo” or brown was the most common self-reported race in Brazil in the year of 2022. We have added a sentence in the methods section including this definition. However, our manuscript does not dive deep into the matter of race, as it was not the main focus of discussion, but rather additional information to understand the distribution of vaccination among different Brazilian regions.

Reference: Censo 2022: pela primeira vez, desde 1991, a maior parte da população do Brasil se declara parda | Agência de Notícias (ibge.gov.br)

The same applies to indigenous population. Usually, Brazilian indigenous people in littoral areas identifies themselves as blacks or “browns??” than Indians to avoid more discrimination, as indigenous people are considered less reliable than other ethnos, where others with high miscegenation and only with a little indigenous background must identify themselves as Indigenous to achieve public benefits.  

Thank you very much for this comment. As vaccination coverage according to race is a secondary point of discussion in our manuscript, we have opted not to address this specific topic in further detail. We have now included the rationale for using these definitions in our manuscript (in agreement with IBGE as mentioned above).

Data are presented in long tables without statistical analysis, resulting in difficult reading and comprehension. Those tables could be easily resumed if variables that are much segmented as age groups could be transported to presentation as figures. Figures are also a problem. Maps without population density are not conclusive. When looking the Brazilian maps, North region is the large but with the smaller population. Maps without adequate information only obscure the data significance.

Revisiting the manuscript, we agree with reviewer 1 regarding this topic. To address this problem, we have decided to relocate tables 2 and 3 as Supplemental material. In addition, we have added information about the populational density of Brazilian regions to facilitate the interpretation of maps. Our discussion now focuses more on the local production of vaccines and how it influenced vaccine availability in the country.

Populational density was included as follows (2010)

  • Brazil: 22,43 inhabitants/km²
  • North: 3,35 inhabitants/km²
  • Northeast: 30,69 inhabitants/km²
  • Southeast: 86,92 inhabitants/km²
  • South: 48,58 inhabitants/km²
  • Center-west: 8,75 inhabitants/km²

Reference: IBGE Censo 2010

The authors must focused their results in less tables and statistical analysis, not looking for the regional or ethnic differences but in local or elsewhere vaccine production, which is much more important for a subcontinental country as Brazil.

In agreement with these suggestions, we have removed two tables and relocated them as supplemental material. In addition, we have further endorsed the implications of local production in vaccine distribution (as we are not able to analyze the local production itself) in the discussion and conclusions.

Reviewer 2 Report

Comments and Suggestions for Authors

Manuscript: Lessons learned from the COVID-19 pandemic

Summary: The authors present data from a nationwide Brazilian study on individuals that were vaccinated in the first 9 months of 2021.  Despite an incredible amount of data in the study, the authors put together a series of analysis that made the data more understandable for the reader.  The data analysis was broken down in various ways to look at different aspects of the population, geographic regions, and vaccine availability.

Reviewer notes and suggestions:

1.      The amount of data presented in the manuscript is very large.  The authors may consider placing some or the bulk of the data in supplemental tables to reduce the size in the main manuscript (just focusing on the summary statistics).  The authors may create a smaller table or figure that summarizes the various vaccine constructs, availability, and immunization recommendations for the reader to refer back to as needed.

2.      The authors provide some explanations for various aspects of the vaccination campaign based on the data.  However, the authors could greatly improve the manuscript by putting a small table or graphic outlining some of these lessons learned both aspects that worked well vs those that did not.

3.      The authors mention that the geographical regions have differences in urbanization and areas that are difficult to reach.  Could the authors expand on this aspect:  What are some differences in infrastructure that led to the disparity between some of these regions?  Any suggestions or ideas of what aspects of the existing infrastructure to prioritize to improve vaccinations in the future?

4.      In the discussion section the authors focus on numerous aspects of the vaccination effort (vaccination rates, demographics, delivery, number of boosters, availability, etc.).  The manuscript could be more impactful if the authors provide some recommendations or ideas in the conclusion section on what to keep in place and what things to improve if this ever happens again. Any suggestions or recommendations for DataSUS and ANVISA? Based on the lessons learned what improvements could be made for vaccinating the population in Brazil?  Other countries?

Author Response

Dear Reviewers, thank you very much for your careful and productive comments regarding our manuscript. Please find a point-by-point answer to your questions and suggestions below. If feel that our answers are still not complete enough, or if there are still open questions, we would be more than willing to discuss these again.

  1. The amount of data presented in the manuscript is very large.  The authors may consider placing some or the bulk of the data in supplemental tables to reduce the size in the main manuscript (just focusing on the summary statistics).  The authors may create a smaller table or figure that summarizes the various vaccine constructs, availability, and immunization recommendations for the reader to refer back to as needed.

We thank reviewer 2 for this relevant suggestion. We have now relocated tables 2 and 3 as supplemental material. In addition, we have created a new table summarizing the lessons learned, divided by successes and challenges of the vaccination campaign.

  1. The authors provide some explanations for various aspects of the vaccination campaign based on the data.  However, the authors could greatly improve the manuscript by putting a small table or graphic outlining some of these lessons learned both aspects that worked well vs those that did not.

We have now added a table summarizing the successes and challenges of the vaccination campaign as a more practical approach to the lessons learned with COVID-19 vaccination in Brazil. We thank the reviewer for this thoughtful suggestion.

  1. The authors mention that the geographical regions have differences in urbanization and areas that are difficult to reach.  Could the authors expand on this aspect:  What are some differences in infrastructure that led to the disparity between some of these regions?  Any suggestions or ideas of what aspects of the existing infrastructure to prioritize to improve vaccinations in the future?

We agree with reviewer 2 in this regard. As this manuscript highlights the pitfalls and successes of vaccination, we have added new information into our discussion that may help to understand why the north and northeast regions may have had lower vaccine coverage based on geographical and infrastructural difficulties, especially regarding the extension of rainforest, challenges in properly storing and transporting biological products, amongst others.

  1. In the discussion section the authors focus on numerous aspects of the vaccination effort (vaccination rates, demographics, delivery, number of boosters, availability, etc.).  The manuscript could be more impactful if the authors provide some recommendations or ideas in the conclusion section on what to keep in place and what things to improve if this ever happens again. Any suggestions or recommendations for DataSUS and ANVISA? Based on the lessons learned what improvements could be made for vaccinating the population in Brazil?  Other countries.

We thank the reviewer for this suggestion. As this manuscript tries to look back at the vaccination strategies and their accomplishments and challenges, it is important to highlight where there is still a place for improvement. With our new table, we address the points that should be further encouraged in new vaccination programs as well as pitfalls that still need strategic improvement.

Round 2

Reviewer 1 Report

Comments and Suggestions for Authors

This manuscript is interesting by the large numbers of vaccinees and the several types of vaccine used and their logistic problems. In the present version is acceptable for publication despite larger tables.